# The Effectiveness of L-arginine in Clinical Conditions Associated with Hypoxia

**DOI:** 10.3390/ijms24098205

**Published:** 2023-05-03

**Authors:** Natalia Kurhaluk

**Affiliations:** Department of Biology, Institute of Biology and Earth Sciences, Pomeranian University in Słupsk, Arciszewski St. 22 B, 76-200 Słupsk, Poland; natalia.kurhaluk@apsl.edu.pl

**Keywords:** effectiveness of L-arginine in therapeutic practice, NO and individual physiological reactivity, favourable effect of L-arginine supplementation, high resistance to hypoxia, NO-dependent biosynthesis pathways

## Abstract

The review summarises the data of the last 50 years on the effectiveness of the amino acid L-arginine in therapeutic practice in conditions accompanied by different-origin hypoxia. The aim of this review was to analyse the literature and our research data on the role of nitric oxide in the modulation of individual physiological reactivity to hypoxia. The review considers the possibility of eliminating methodological conflicts in the case of L-arginine, which can be solved by taking into account individual physiological reactivity (or the hypoxia resistance factor). Considerable attention is paid to genetic and epigenetic mechanisms of adaptation to hypoxia and conditions of adaptation in different models. The article presents data on the clinical effectiveness of L-arginine in cardiovascular system diseases (hypertension, atherosclerosis, coronary heart disease, etc.) and stress disorders associated with these diseases. The review presents a generalised analysis of techniques, data on L-arginine use by athletes, and the ambiguous role of NO in the physiology and pathology of hypoxic states shown via nitric oxide synthesis. Data on the protective effects of adaptation in the formation of individual high reactivity in sportsmen are demonstrated. The review demonstrates a favourable effect of supplementation with L-arginine and its application depending on mitochondrial oxidative phosphorylation processes and biochemical indices in groups of individuals with low and high capacity of adaptation to hypoxia. In individuals with high initial anti-hypoxic reserves, these favourable effects are achieved by the blockade of NO-dependent biosynthesis pathways. Therefore, the methodological tasks of physiological experiments and the therapeutic consequences of treatment should include a component depending on the basic level of physiological reactivity.

## 1. Introduction

In recent decades, the processes of urbanisation and industrialisation, as well as other changes in human lifestyles, have significantly increased the development of various diseases, among which cardiovascular pathology has become the leading cause of death in industrialised countries. The search for new effective methods of preventing and treating these diseases is therefore extremely important [1]. One such effective approach is the stimulation of the body’s natural defences through adaptation or combined methods of adaptation and an appropriate diet. The contribution of active longevity determinants to human health includes several components [2] that may differ from the main literature reports from recent decades [3].

Reports presenting pronounced therapeutic effects that have been confirmed in evidence-based medicine deserve special attention. First of all, attention is drawn to studies of genetically homogenous populations of long-livers who live only in “mountain longevity zones” and have developed over centuries as a result of their long-term residence in these specific geographical regions [4]. The available data indicate that residency at higher altitudes is associated with lower mortality from cardiovascular diseases, stroke, and certain types of cancer. Scientists have pinpointed one reason why people living in isolated villages in Greece may enjoy long and healthy lives. They found a new genetic variant, common among villagers, which appears to protect the heart by lowering the levels of “bad” fats and cholesterol. Despite a diet rich in animal fat, the people of Mylopotamos in northern Crete do not suffer from cardiovascular disease. It was shown that the genetic variant R19X was far more common in this population than in other European populations—2 per cent compared to just 0.5 per cent [5].

The findings described above are associated with lipid metabolism and the risk of cardiovascular disease [6]. The benefits of the Mediterranean diet are well known, but scientists have discovered that the long and healthy lives of Greeks living in isolated mountain villages may not necessarily be related to their diet [4]. It should be noted that the effects of effective longevity without cardiovascular risk are considered not only in connection with the use of a Mediterranean diet but also with the conditions of moderate hypoxia in the place where these people live [7,8]. Thus, the effects of the phenomenon of hypoxia and longevity have always been the focus of research [9,10]. It should be noted that the effects of genetics are commensurate with the effects of nutrition, amounting to 20–25% since food must constantly be supplied to the organism. Importantly, the quantity and quality of food can also contribute largely to active longevity, and there are no dramatic qualitative differences in the dietary patterns of long-livers compared to the general population, such as the Mediterranean diet users mentioned above.

In the last 40 years, studies of the metabolism of the amino acid L-arginine have been important for evidence-based medicine [11,12]. The physiological need of tissues and organs of most mammals for arginine is satisfied by its endogenous synthesis and/or intake with food; however, this amino acid becomes essential for young individuals and adults in conditions of stress or illness [13]. Arginine serves as a necessary precursor for the synthesis of proteins and many biologically important molecules such as ornithine, proline, polyamines, creatine, and agmatine. The main role of arginine in the human body is to be a substrate for the synthesis of nitric oxide (NO), as shown earlier by Zembowicz [14]. A wide range of physiological and pathological mechanisms of mitigation of many body conditions has been the subject of research over the past half a century [15]. The metabolic NO-dependent pathways and potential clinical use of L-arginine are shown in Figure 1 and Figure 2.

The relationships of oxygen with the synthesis of amino acids and the effects of hypoxia in the body have been shown; therefore, the therapeutic use of the amino acid itself and its precursors in sports deserves attention [16]. However, the conflicting data on the use of L-arginine during extreme hypoxic loads, especially in sports, raise a question about the role of this important supplement in the functioning of the body in these conditions [17,18]. Is it possible to reduce or successfully circumvent the genetic factor by considering a diet, appropriate exercise, and various stress-coping practices as essential? This factor changes from positive to negative for residents of large cities located high above sea level.

It is known that the regulation of metabolism in individuals with different (high or low) baseline resistance to hypoxia is determined by the cholinergic mechanisms of regulation due to the increased importance of acetylcholine in the course of more energy-efficient reactions [19]. The high efficiency of ATP production processes in energy transformations in conditions of oxygen deficiency is associated with the increased role of cGMP, which can act as a negative feedback effector [20]. It is known that an increase in the content of cGMP can inhibit the intensity of redox processes involving nitric oxide and thereby reduce the need for tissue oxygen [21]. These studies were focused on the functioning of NO-synthase systems and the level of nitric oxide produced in these conditions. Such effects on the functioning of mitochondrial processes, in particular, the conjugation of respiration and phosphorylation processes, their efficiency, and the intensity of lipoperoxidation processes depend on the basic level of adaptation and resistance to hypoxia, as convincingly proved by many authors and our previous studies [19,22].

The research on the antioxidant defence system and lipid peroxidation processes in animals that have different sensitivity to hypoxia provides important information and can be extrapolated to humans. Data presented by a number of authors show the different activity of enzymes and metabolites in control conditions related to the different abilities to inactivate reactive oxygen species (ROS) under stress accompanied by hypoxic processes [23]. This characterises the reserve compensatory capabilities of the organism exposed to adverse environmental factors. They were found to be better expressed in rats with high resistance to hypoxia, which was also manifested in stress-limiting functions under high loads in these individuals. The aim of this work was, therefore, to review the literature published over the last 50 years and the results of our research on various aspects of the application and supplementation of L-arginine in athletes in order to eliminate the methodological difficulties resulting from differences in data obtained in therapeutic experiments.

## 2. Cardiovascular Disease, L-arginine, and NO

Vascular endothelial dysfunction occurring in many diseases (atherosclerosis, hypercholesterolaemia, hypertension, and diabetes) is characterised by impaired secretion of protective endothelial mediators, including endothelial vasodilator factor (EDRF), recognised in the 1980s and now identified with the nitric oxide (NO) molecule [18]. It was also discovered that its only substrate in humans is the well-known amino acid L-arginine. For the discovery of the role of NO as a signalling molecule in the cardiovascular system, R. Furchgott, L. Ignarro, and F. Murad were awarded the Nobel Prize in Medicine and Physiology in 1998 [19].

NO serves many functions in the physiology and pathology of the organism [24,25]. The pathology of many diseases is ameliorated in clinical practice by supplementation of L-arginine [26] (Rashid et al., 2020). Selected examples are shown in Table 1. Pathological changes in the cardiovascular system, known as vascular endothelial dysfunctions, are often associated with a reduction in the synthesis of NO [27]. Oxidative stress in endothelial cells is associated with vascular endothelial dysfunctions and reduced NO synthesis, which are related to the aetiology of many pathological changes in the cardiovascular system [18].

It is now well established that NO plays a key role in microcirculatory disorders in sepsis [47]. It is associated with the development of septic shock when a large number of microbes circulating in the blood sharply activates the synthesis of NO in the endothelium. This leads to prolonged and severe dilatation of small blood vessels and, consequently to a significant decrease in blood pressure. The overproduction of NO in the vascular wall due to a sharp drop in vascular resistance causes potentially lethal hypotension in septic shock and multiple organic lesions in humans, which can lead to vascular collapse [48].

Oxidative stress in blood vessel wall cells alters blood flow regulation and inhibits platelet adhesion and aggregation, leukocyte adhesion, and cell growth. It also promotes Ca^2+^ entry into vascular smooth muscle cells, stimulating changes that lead to the development of atherosclerosis and hypertensive disease [30]. Examples of cardiovascular diseases whose pathogenesis is likely to be related to disturbances in NO synthesis or action are hypertensive disease and atherosclerosis [27,49]. Therefore, the administration of compounds from which NO can be formed, e.g., NO substitutes (nitroglycerine and organic nitrates or arginine), to hypertensive patients and supplementation with antioxidants may be important in the treatment of hypertension due to the protective effect of scavenging/inactivating oxygen free radicals [50].

A parallel pilot study evaluated the 14-day effect of oral L-arginine supplementation (at a dose of 6 g per day) in 22 patients with chronic heart failure treated with an angiotensin-converting enzyme inhibitor, a diuretic, and digoxin. Initially, a clinical improvement reflected by a shift of the patients to classes with better physical endurance, an increase in the cardiac output fraction, an increase in serum NO and cGMP concentrations, and a decrease in the magnitude of peripheral resistance were earlier reported [51]. Supplementation with L-arginine was reported to result in an increase in nitric oxide synthesis [52].

## 3. L-arginine in Therapeutic Applications

The role of NO in the maintenance of vascular homeostasis is limited to the regulation of vascular tone, proliferation, and apoptosis, as well as the regulation of oxidative processes. In addition, NO has inherent angioprotective properties. It is also responsible for such anti-inflammatory effects as the inhibition of the expression of cellular adhesion molecules [53]. Potential indications for the clinical use of L-arginine include spontaneous hypertension [27], pregnancy-induced hypertension [18,29,31], pulmonary hypertension [26,30,31], lower limb arteriosclerosis [32], Raynaud’s phenomenon [34], hypercholesterolaemia [36,37], stable ischaemic heart disease [38], endothelial function [40,41], circulatory insufficiency in loss- and gain-of-function models for eNOS in Cre-induced gene inactivation [54], regulation of blood pressure and microcirculation [42], glaucoma [41], diabetes mellitus [39,42], aerobic training in chronic heart failure [45], chronic renal failure [55], prevention of coronary artery restenosis [49], and prevention of stroke and thrombotic vascular incidents [56].

L-citrulline, a natural precursor of L-arginine, is more bioavailable than L-arginine because it avoids hepatic first-pass metabolism and has a longer circulation time. Arginine/citrulline has immense therapeutic potential in some life-threatening diseases in patients of different ages [26]. Convincing results were obtained using various ways of application of L-arginine, and positive effects of this amino acid administered in medical practice to reduce its deficiency or correct pathological conditions in the cardiovascular system were reported (Figure 2).

## 4. Hypoxic Conditions and L-arginine

The evolutionarily oldest way of adaptation of the organism is associated with the ability to tolerate different levels of oxygen deprivation and the effect of other adverse factors, which can be the basis for the correction of many pathological conditions [50]. The main mechanism of the harmful hypoxia effect is the disruption of oxidation processes in conditions of oxygen deficiency through uncoupling of respiration and oxidation processes and switching the metabolism to the glycolytic pathway, which, however, does not provide sufficient energy production. This is accompanied by a decrease in the content of macroergs, accumulation of non-oxidised products, and intensification of free radical formation [57]. The generation of ROS is an essential moment of hypoxic cell damage [50]. Finally, there are pathophysiological disorders of hemodynamics and microcirculation, loss of enzymes by cells, and damage to membrane structures and organelles, which inevitably activates reactive oxygen species (ROS) [58]. Therefore, the reaction to the effects of different-origin hypoxia (intense physical activity, acute stress, etc.) reflects the reserve compensatory capabilities of the body under the influence of extreme environmental factors [22,23,57]. Figure 3 presents the main mechanisms of responses to oxygen deficiency.

Moderate hypoxia, corresponding to living at moderate altitudes, and persistent physical stress accompanied by tissue hypoxia/reoxygenation are regarded to be stressors. Next, preconditioning mechanisms develop [59], which enhance the level of individual resistance or resistance to the hypoxic factor. Currently, close attention is being paid to both pharmacological and non-drug methods of stimulation of the adaptive potential of the organism. They include various types of adaptation to, e.g., emotional stress, physical and hypoxic loads, and changes in environmental temperature. This is particularly important for modern man in the intensely changing world, given the availability of stress-increasing information, sedentary lifestyle, unbalanced diet, and accompanying cardiovascular diseases [60].

It should be noted that the NO-synthase mechanism involves the synthesis of NO from L-arginine in the presence of oxygen [61]. It has been shown that hypoxia reduces the transport and synthesis of this amino acid [57]. Therefore, under functional loads and different-origin hypoxia, another mechanism of NO synthesis associated with the reduction of NO^2−^ anions to NO can be activated. Therefore, the sequence of metabolic transformations: L-arginine → NO → NO^2−^/NO^3−^ → NO forms a cycle called the NO cycle [62]. The creation of temporary hypoxic conditions by switching energy metabolism from aerobic to anaerobic processes helps to maintain cellular and tissue homeostasis. Perhaps, such restructuring of the energy pathways of metabolic supply is associated with different inhibition of the NADH- and FAD-dependent parts of the respiratory chain by nitric oxide [63] and is aimed at preventing energy deficiency under acute hypoxia. Depending on many factors, NO plays both prooxidant and antioxidant roles in vitro [64]. Other factors in the pathogenesis of mitochondrial damage and respiratory chain dysfunction in hypoxia and ischemia are the products of free radical reactions [65].

It is known that many pathological conditions accompanied by hypoxia are characterised by a decrease in the capacity of nitric oxide generation systems [66]. At the same time, shocks of different origins are characterised by hyperproduction of NO, which leads to impaired vasoconstrictor responses [28,29,67]. Thus, balancing the production of NO in the body both by adaptation and under the influence of pharmacological drugs can be an effective means of preventing and treating diseases associated with both hypo- and hyper-production of NO and the total antioxidant system [18].

## 5. Supplementation of L-arginine

Arginine supplied with food is released from proteins during the digestive process and absorbed in the small intestine [68]. The main site of its absorption is the ileum and jejunum. However, due to the high arginase activity in the small intestine, approximately 40% of the arginine from food is degraded here, and the remaining amount goes into the portal vein. It is believed that about 50% of dietary arginine enters the circulatory system [13], 5–15% of arginine in adult blood plasma is synthesised endogenously from citrulline. Total plasma arginine concentrations in humans and animals are in the range of 95–250 μmol/L and depend on the developmental stage and nutritional status of the organism, as this team established earlier [69].

Given the ability of arginine to lower such parameters as glucose, homocysteine, triglycerides, and fatty acids, its supplementation is a prospect in the treatment of obesity and other disorders comprised associated with metabolic syndrome [70]. The demand for this amino acid increases during parenteral nutrition, in certain physiological states (stress, illness), and in pathophysiological conditions associated with disease, injury, or significant strain on the body. Immediately after release from proteins, 50% of the arginine supplied with food enters the blood. Arginase breaks down 40% of arginine in the small intestine [71]. High arginine content is found in meat, seafood, nuts, seeds, and groats [72]. Arginine from plant proteins is more bioavailable to the human body than from animal sources. This has to do with the ratio of arginine to lysine, i.e., a competing amino acid in the absorption process, which is higher in plant products than in animal ones [73,74,75].

The effect of intravenous arginine infusion has been shown to increase growth hormone (GH) secretion by the anterior pituitary lobe, which is related to the inhibitory effect of this amino acid on endogenous somatostatin secretion. GH, especially in higher amounts, can negatively affect the human body through mitogenic effects, increasing the risk of cancer [76]. GH has also been found to reduce carbohydrate utilisation by decreasing glucose uptake, which can hinder the delivery of energy, particularly in endurance sports [77,78,79]. Collier et al. [80] showed a higher maximum GH concentration in a group taking 9 g Arg than in those supplemented with 5 g Arg and placebo. In addition, gastrointestinal problems were found with 13 g Arg supplementation, with no significant effect on GH levels. Arginine appears to have both beneficial and detrimental effects on kidney function. However, adverse effects are unlikely to occur with routine doses (from 3 to >100 g/day), as shown by the authors [81]. Thus, the complex network of pathways of L-arginine metabolism in the body and its supplementation are important dietary sources used for the prevention of many pathological conditions [52,82]. Therefore, the addition of arginine to the diet when its intake is insufficient may be necessary to maintain optimum homeostasis of this amino acid in certain physiological and pathological conditions, depending on the species and age.

## 6. L-arginine Paradox

Some of the current most popular supplements are so-called pre-training nitric oxide (NO) boosters [16]. They contain a wide variety of ingredients, but the most important one is L-arginine, mainly because L-arginine is a precursor of NO, and NO is a powerful vasodilator, which is able to increase blood flow in muscles during training, improve athletic performance, and speed up recovery. Popular pre-exercise supplements that increase NO levels rely on the L-arginine-NO pathway and a number of theoretical assumptions [83]. The conversion of arginine to nitric oxide (NO) is catalysed by nitric oxide synthase and asymmetric dimethylarginine (ADMA). It is suggested that plasma levels of ADMA can be reduced by pharmacotherapy techniques. It is considered that ADMA plays an important role in the regulation of vascular tone by acting as an endogenous inhibitor of nitric oxide synthase [84]. By inhibiting NO synthesis, ADMA decreases vascular elasticity, increases vascular resistance, and limits blood flow to organs and tissues. In addition, increased blood levels of ADMA contribute to the development of atherosclerosis by eliminating the vasoprotective effects of NO. Thus, increased plasma ADMA levels may accelerate the development of atherosclerosis and increase the risk of cardiovascular disease and its complications [85,86].

One of the causes of the inconsistent results of studies on arginine supplements is that they have used different routes of administration (oral and intravenous), different forms of L-arginine, different exercise test protocols, different subjects (trained and untrained, young and old), and shakes containing various additional substances (e.g., creatine, beta-alanine, caffeine, carbs, etc.) that can actually increase athletic performance [68]. However, although taking L-arginine supplements does have positive effects on people with various painful conditions, studies of healthy people cannot unequivocally support the advertising claims about the effects of L-arginine supplements and nitric oxide boosters [69]. Why?

The main compound that controls the arginine-NO pathway is the eNOS enzyme (endothelial nitric oxide synthase). The concentration of its substrate is a factor affecting the rate of reactions catalysed by this enzyme. L-arginine is a substrate for NOS, which converts it into NO [87,88]. For eNOS, the Michaelis-Menten constant (Km) (i.e., the substrate concentration at which an enzyme reaction rate of 50 per cent of the maximum is achieved) is lower than the L-arginine concentration in the blood of both healthy and sick people [52].

This means that normal non-enhanced L-arginine levels in the blood are already high enough to saturate endothelial NOS. When the enzyme is already saturated with the substrate (in this case, L-arginine), a higher volume of substrate will have no effect on the response. Therefore, taking L-arginine (even in high doses) will not enhance NO production because L-arginine is not a rate-limiting factor for eNOS. However, in the presence of clinical conditions, e.g., high blood pressure, elevated cholesterol, heart disease, insulin resistance, or diabetes, and in the elderly, taking L-arginine as a dietary supplement has positive effects, probably as a result of increased NO production [89].

Thus, treatment of NO-deficiency disorders can also be achieved by administration of the substrate from which it is formed, namely L-arginine. The sufficient activity of the constitutive NO synthase in tissues is a prerequisite for effective treatment. The administration of excess substrate does not lead to the production of excessive amounts of NO [90], as these can only be formed if the enzyme is induced, and the excess substrate does not cause this process.

## 7. L-arginine and Sport

Is arginine supplementation an effective method of supporting exercise capacity in sports? The literature in recent years has been ambiguous on this point [87]. A major advantage of arginine is the safety of its use. Arginine, which occurs as a pure L-arginine preparation or in combination with other amino acids and their derivatives, is currently one of the most common supplements used by athletes. On average, 3–6 g of arginine is consumed per day with both animal and plant protein products, which are rich sources of this amino acid.

Actually, most of these studies demonstrate the positive effects of arginine supply on increased physical performance and beneficial regulation of endocrine parameters [91]. Noteworthy, results obtained in highly trained healthy athletes are inconclusive and often demonstrate no effect of arginine preparations on power, strength and muscle mass, maximum oxygen uptake, and growth hormone and nitric oxide concentrations [87]. A number of clinical studies involving convalescents and patients with cardiovascular disease have shown an association of arginine supply with increased physical performance and beneficial regulation of endocrine parameters [16,42,92,93].

One of the most important metabolites of L-arginine with documented beneficial effects on increased exercise capacity is creatine, which is synthesised mainly in the kidney and in smaller amounts in the liver and pancreas [94]. Creatine plays a key role in muscle energetics [95]. The metabolism of arginine leads to the production of approximately 1–2 g of creatine per day, indicating a need for additional supply from the diet [81]. Approximately 90–95% of the body’s creatine is found in skeletal muscles, with 2/3 of this pool being phosphocreatine and the remainder being free creatine. Due to its potentially ergogenic effects through stimulation of nitric oxide, growth hormone, and creatine synthesis, arginine is often supplemented in athletes [81]. An increase in the concentration of PCr contributes to faster resynthesis of ATP, i.e., an essential energy source for proper muscle contraction, especially during supramaximal efforts [96]. In addition, creatine has been observed to improve aerobic metabolism and reduce lactate acidosis. In the case of athletes, the above-mentioned effects contributed to an increase in training loads and a change in body mass and composition (an increase in muscle mass) [97]. The effectiveness of creatine has been confirmed in endurance and strength sports [98].

Earlier, Evans et al. [99] and Collier et al. [80] have found that the optimal amount of arginine that increases blood arginine levels, changed glycemic profile [100] and does not cause gastric problems is 9 g. The efficacy of arginine was demonstrated during supplementation with 12 g/day of L-arginine α-ketoglutarate for eight weeks [101]. In the treatment group, significant increases in maximal power in the bench press, maximal power determined by the Wingate test, and plasma arginine concentrations were observed compared to the placebo group. However, no differences in the body composition were found between the study groups. A group of strength-training athletes taking 12 g/day of L-arginine α-ketoglutarate for eight weeks showed a significant increase in haemoglobin levels and blood arginine concentrations [101]. However, there were no significant differences between the L-arginine α-ketoglutarate intake group in the oxygen uptake, minute ventilation, and gas exchange ratio. In a four-week study of endurance athletes taking arginine and aspartate, Abel et al. [102] found no significant differences between groups in VO_2_ max achieved and time to exhaustion during the test. Furthermore, no effect of the supplementation with the tested preparation on changes in the levels of growth hormone, testosterone, cortisol, glucagon, and lactate was observed. A more recent systematic review by other authors on the effects of L-arginine supplementation on VO_2_ max in healthy subjects, without significant heterogeneity in the meta-analysis, concluded that these supplements might increase VO_2_ max in healthy subjects [103]. Studies by other authors on athletes [104,105,106] do not support the use of L-citrulline, L-arginine, or Beetroot juice supplements as ergogenic agents to improve physical performance. Therefore, now the theoretical justification for the use of L-arginine in highly trained athletes was significantly revised, and even abandonment of the use of this dietary supplement was proposed and strongly justified. It is possible that this phenomenon may be related primarily to the adaptation to high hypoxic loads and the functioning of the NO system under intensive training. In turn, the NO system is activated in this group of individuals with high physiological reactivity.

## 8. Individual Physiological Reactivity and NO

The NO system function in conditions of L-arginine supplementation needs to be preliminarily evaluated via the profile of individual resistance to hypoxia in light of the statistical results presented in the literature. It is believed that this can be done in a statistically valid way if the analysed parameters of animals (as well as groups of analysed volunteers) exhibit normal distribution and the studied sample can be correlated with the general population of individuals to be analysed, called the population transfer [107,108,109]. At the same time, it is known that, in laboratory animals and humans, there are always differences in individual sensitivity to hypoxic factor effects, which often correlates with resistance to dynamic stress and stress tolerance [19,109,110]. Therefore, extreme hypoxia itself is an additional factor whose effect deviates from the statistical characteristics of functional indicators of cell energy supply and the level of autonomic nervous system functioning from normal statistical distribution [111]. Therefore, the resolution of the methodological challenges requires taking these factors into consideration when analysing the effects of pharmacological preparations and dietary supplements with anti-hypoxic, antioxidant, and other properties on specialised biological models.

In contrast to studies in patients, the results obtained in highly trained healthy athletes are inconclusive. It seems that the great popularity of supplements is related to extensive advertising campaigns by sports nutrition manufacturers, who suggest that taking arginine leads to an increase in exercise capacity by stimulating the synthesis of nitric oxide, growth hormone, and creatine. It is worth mentioning that, in order to prove the effectiveness of these supplements, the results of clinical trials involving, e.g., people with cardiovascular disease and in recovery are often used [4,82]. These results often demonstrate no effect of L-arginine preparations on aerobic metabolism in athletes and growth hormone levels, as previously shown in some studies [87].

In their studies on athletes using L-arginine derivatives or its precursors, authors [101,102,104,105,106] found no significant differences in athletic performance. In this situation, the need for further research into the advisability of increasing arginine supply in athletes of selected sports is justified; however, these studies must follow appropriate procedures, including double-blind crossover studies. Additionally, in this situation, it will be justified to conduct further research into the advisability of increasing arginine supply in athletes and non-athletes, taking into account studies of individual physiological reactivity characteristics. These characteristics can be explained by the body’s individual response to acute oxygen deprivation, i.e., hypoxia.

This concept can also be used in relation to certain types of diets containing L-arginine as a precursor to nitric oxide. This is important given the effects of heavy metals in individuals with different baseline levels of hypoxia resistance, as we demonstrated in our study of lead toxicity and cobalt effects in these animals in stress conditions [110]. It is possible that the recommended diet containing L-arginine as a precursor of NO may help to reinforce the positive and creative role of stress, as proposed by H. Selye, who believed that there would be no life without stress. This is related to the desire to move beyond the pathological model of stress to research concepts that can shed light on the mechanisms of eustress. Such effects were observed in animals, i.e., during acute hypoxia-induced activation of mitochondrial oxidation, the parameters of mitochondrial respiration decreased using the main oxidation substrates in the respiratory chain in the animals after the adaptation in an interval mode [112].

These trends can be linked with changes in the development of specific training elements when the role of the nitric oxide system increases. Our studies and results reported by other authors have convincingly shown that interval hypoxia used as a method of correction of many pathological conditions, preventing activation of free radical oxidation processes in tissues and blood under the influence of fasting hypoxia, determines the increase in the efficiency of energy reactions in cell mitochondria. Interval hypoxia used as an effective training method is mediated by increasing the role of NO in adaptation mechanisms [113]. Importantly, it is accompanied by an increase in the role of the antioxidant regulatory effect, the antioxidant protection system, and the intensification of lipoperoxidation processes, which is considered by some authors as good stress (eustress); it provides energy, strengthens the heart muscle, and enhances cognitive function [114]. It is important that the favourable effect of L-arginine supplementation was noted only in a group with low adaptive capacity to hypoxia and individuals with high resistance to hypoxia in stress conditions exhibited blockade of NO-dependent biosynthesis pathways. Therefore, the methodological tasks of physiological experiments and the therapeutic consequences of treatment should include a component depending on the basic level of physiological reactivity.

## 9. Resistance to Hypoxia, Mitochondrial Energy Support, and L-arginine

The regulation of energy homeostasis is largely dependent on the functioning of the Krebs cycle and the associated energy supply of the cell in hypoxic conditions [115], which accompany pathological conditions of different origins [58]. An important role of NO in the regulation of oxygen-dependent processes in the cell during acute hypoxia has been shown [116,117]. Our systematic study of the effect of the main intermediates of the Krebs cycle with the use of succinic and α-ketoglutaric acids on the metabolism of nitric oxide in the blood and tissues of rats under acute hypoxia has demonstrated that α-ketoglutarate has a more pronounced effect than succinate on the nitrite anion content in the liver under acute hypoxia. Therefore, reducing the intensity of lipid peroxidation in these conditions was an important factor in the protective effect of sodium α-ketoglutarate [110,112].

This finding allowed establishing a connection between the processes of energy supply, the intensity of lipoperoxidation, the activity of antioxidant defence enzymes, and NO production in mitochondria. The results obtained show that, depending on the initial level of physiological reactivity, i.e., the predominance of initial high or low resistance to the influence of the hypoxic factor [19,110], there is a modification of calcium accumulation and energy supply processes mediated by the influence of the nitric oxide system, as shown in Figure 4. In the experiment, we demonstrated the influence of exogenous regulators of nitric oxide synthesis, such as L-arginine or NO synthase inhibitor L-NNA. It was found that the activation of NO synthesis using its precursor increases the adaptive potential of organisms with a low resistance to hypoxia; in animals with high resistance, the correction of energy supply processes under stress is carried out under the influence of the NO synthase inhibitor (Figure 5).

Previously, it has been convincingly shown that targeted correction of the NO generation system can be a promising tool for the prevention of pathological conditions accompanied by various levels of tissue hypoxia, which can cause stress in mitochondrial energy supply processes [118]. Such a condition often causes electron escape from the mitochondrial respiratory chain and activation of lipid peroxidation processes, which has been convincingly shown in recent works [119]. No less importantly, these processes must necessarily take into account the individual state of physiological reactivity [19]. This refers to individuals with enhanced activity of cholinergic regulatory mechanisms that already have the potential for adaptive capacity, e.g., rats with high resistance to hypoxia and guinea pigs [120]. Thus, convincing results of the connection between cholinergic mechanisms and the L-arginine-NO system were obtained since one of the mechanisms for the synthesis of nitric oxide is carried out through cholinergic receptors, which has its own evolutionary history. The study conducted by Donald et al. [121] presented a schematic representation of signalling pathways leading to the relaxation of vascular smooth muscle in mammals, which may be important as a tool for evolutionary changes. The binding of the acetylcholine ligand to its receptor on the endothelial cell increases intracellular calcium levels, which triggers four possible pathways that lead to vasodilation. Shepherd et al. [122] discussed the evolution of nitric oxide signalling in vertebrate blood vessels and therapeutic applications.

The use of a natural model that allows investigating the genetically determined mechanisms of increased or decreased resistance of the organism in conditions of insufficient oxygen supply to cells may help to purposefully elucidate the pharmacological and natural means of combating oxygen deficiency, which accompanies many pathologies [22]. The natural model of different resistance to hypoxia determines the final effects of supplementation, as demonstrated in this review in the case of L-arginine and the nitric oxide system associated with this amino acid [123]. The ability of NO to reduce respiration and oxidative phosphorylation in mitochondria in various tissues has been shown by a number of authors [63]. These effects were explained by the ability of NO to bind to cytochrome c oxidase due to its higher affinity for this compound compared to oxygen. Such physiological inhibition is reversible up to certain limits. Only at an excessive concentration is NO able to interact with superoxide (O^2−^) formed in the mitochondrial respiratory chain to form peroxynitrite. The latter blocks the respiratory chain in the area of I-III complexes. Such inhibition is irreversible and plays an important role in cell cytostasis [21].

## 10. L-arginine, Hypoxia, and Methodological Challenge

The large body of available experimental and theoretical data allows the nitric oxide generation system to be considered a stress-limiting system [124]. L-arginine and different hypoxia-induced animal models are shown in Table 2. They indicate that NO is a universal factor in the regulation of physiological systems and the genetic apparatus of cells and plays an important role in the mechanism of stress response and adaptation to stress. The hypothesis that acetylcholine activates the part of oxidative processes which is functionally unrelated to oxidative phosphorylation makes it possible to switch cellular respiration to the nitrate-nitrite component under hypoxia [58], which contributes to the survival of animals under in and other extreme conditions. Thus, the functional state of acetylcholine receptors [12,125] under oxygen deficiency determines metabolic cellular and mitochondrial rearrangements not only for the acetylcholine system but also for NO, which exerts its effects through these structures, as shown in some studies [118]. On the other hand, the volume of NO increases significantly during adaptation to hypoxia in parallel with an increase in cholinergic regulatory effects [112,126].

There are differences in individual sensitivity to the hypoxic agent in each population of both laboratory animals and humans, which often correlates with resistance to dynamic loading and endurance to stress and other factors [110]. Therefore, acute hypoxia itself is an additional factor whose effect deviates from a normal statistical distribution of the characteristics of functional indices of cellular energy supply and the level of autonomic nervous system function [22]. It is important that the solution and elimination of this methodological conflict, consisting of different effects of responses of biological systems to the same factor, requires first of all considering the individual characteristics of the physiology of action of hypoxic factors. This approach increases with a thorough analysis of the impact of pharmacological preparations and dietary supplements exerting their beneficial effects on specialised biological models.

Two approaches have traditionally been used to assess the activity of anti-hypoxia treatments. The first approach is used to evaluate a single administration of a drug when group average values of individual resistance to hypoxia are assessed. The second approach to the course of drug administration is a change in the population structure of resistance, i.e., a change in the proportion of individuals with low and high resistance in the total population. The relevance and validity of this assessment are demonstrated by a number of studies on the evaluation of cardiovascular functional characteristics and the number of myocardial infarctions in inhabitants of certain mountainous areas without the influence of diet in these conditions [4].

We showed that the effect of L-arginine was accompanied by increasing dynamic endurance of physical stress in rats with a high and low resistance to hypoxic stimuli. However, the values of the percentage rise were different because the baseline of these processes was different for the two groups. Previously, we observed that L-arginine and alpha-ketoglutarate significantly increased dynamic endurance in the group of animals with low resistance to the hypoxic factor. Cellular mechanisms of low and high resistance to hypoxia are presented in Figure 5. The effects of NO can be regarded as the leading mechanism of adaptation to the VO_2_ max for L-arginine supplementation only in athletes that start training or those that are not athletic in exercise load conditions. Experienced athletes, after a cycle of dynamic, intensive training, do not have such effects of L-arginine supplementation, which has been repeatedly reported [126]. The fact that L-arginine and alpha-ketoglutarate effects disappear under the influence of the nitric oxide synthase inhibitor L-NNA may indicate the interdependence of these processes.

The effects of L-arginine exposure in animals with high physiological reactivity activated via the NO-ergic link of regulation are manifested in conditions of predominant oxidation of alpha-ketoglutarate, exerting unidirectional effects on mitochondrial respiration. These effects may be associated with the impact of NO on different parts of the mitochondrial respiratory chain and reverse interaction with cytochrome c oxidase and I–III mitochondrial enzyme complexes [142]. It is also possible that the effects mediated by the hypoxic factor are exerted through compensatory metabolic flows, which allows the preservation of the energy-synthesising function of the cytochrome c site under stress [21].

In animals that are resistant to stress factors, the main physiological effect of acetylcholine and NO stimulation is to ensure economic oxygen uptake and utilisation by tissues, as shown in Figure 5 and Figure 6. In animals with a low resistance to hypoxia, depending on the intensity of the factor, extraordinary activation of the sympathoadrenal system is often observed, which is accompanied by activation of free radical oxidation. Thus, it is possible to propose a general concept of regulation of mitochondrial functions and the whole cell not only by macroergic compounds but also by coenzymes, metabolites, hormones, and secondary mediators, among which the nitric oxide system has an important regulatory role since it is directly or indirectly involved in all these processes. The reciprocity of these systems is necessary for the regulation of oxygen consumption by cells and its effective use for the synthesis of ATP and other macroergs, depending on the basic physiological and functional state of the organism, and can be subject to targeted correction through supplementation. The establishment of the role of L-arginine supplementation as an endogenous vasoactive factor stimulating NO production [68] outlines the strategy of methodological approaches with an initial consideration of the body’s tolerance of hypoxia and other extreme factors. This methodological solution may be effective in preventing undesirable or toxic effects of known supplements that attract attention to their clinical outcomes. Therefore, the methodological tasks of each physiological experiment and therapeutic study should include a constituent depending on the basic level of physiological reactivity.

## 11. Conclusions

From year to year, the possibilities of clinical use of L-arginine in the treatment of various diseases and in the prevention of civilisation diseases are expanding. The role of arginine as a supplement for athletes is increasing. Results of many clinical trials indicate that arginine can be used both in therapy and for the prevention of many pathologies. The use of even high doses does not show side effects. In light of the research on the role of arginine in cardiovascular disorders, it seems that arginine supplementation may be a potential strategy in the prevention of cardiovascular diseases. However, the dissemination of this strategy requires a final explanation of the mechanism(s) of arginine action in the cardiovascular system, depending on the individual characteristics of NO production in the body caused by the hypoxia factor and its tolerance, and this requires intensive and multi-centre clinical trials, which offer a chance to obtain a safe drug with great therapeutic potential. Therefore, the importance of this review for future perspectives on the methodological challenges of physiological experimentation and the therapeutic implications of treatment consists in including a component that depends on the baseline level of physiological reactivity. 

## Figures and Tables

**Figure 1 ijms-24-08205-f001:**
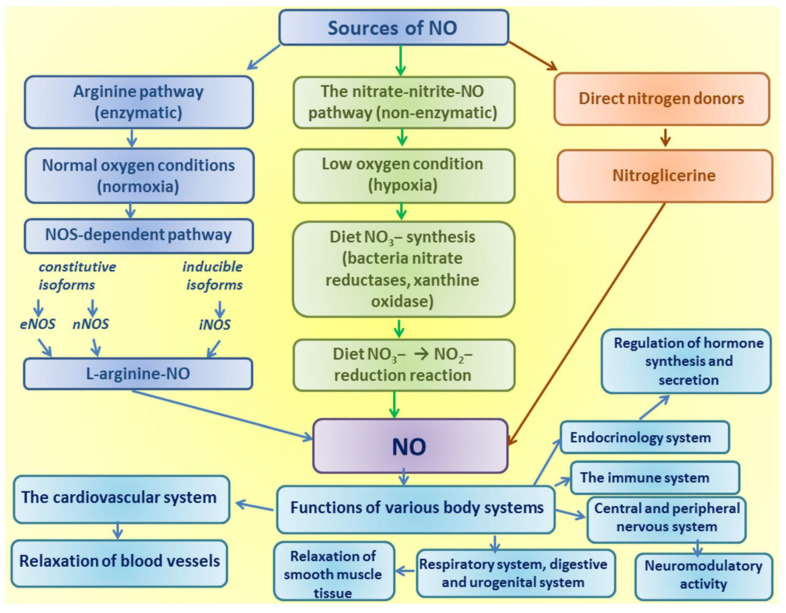
NO-dependent metabolic pathways.

**Figure 2 ijms-24-08205-f002:**
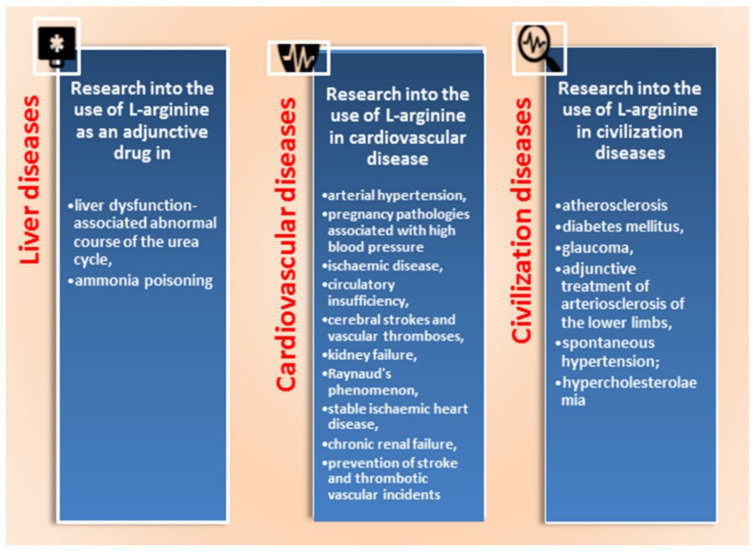
L-arginine in therapeutic applications. Potential directions for the clinical use of L-arginine (detail shown in the text).

**Figure 3 ijms-24-08205-f003:**
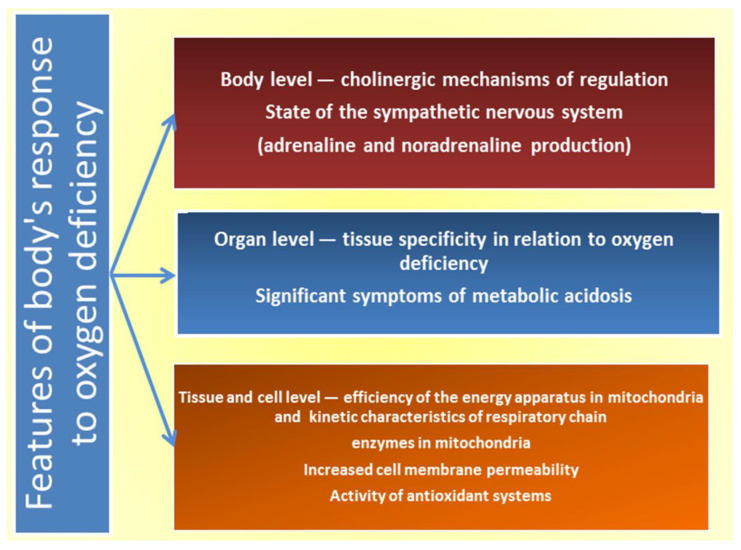
Mechanism of responses to oxygen deficiency.

**Figure 4 ijms-24-08205-f004:**
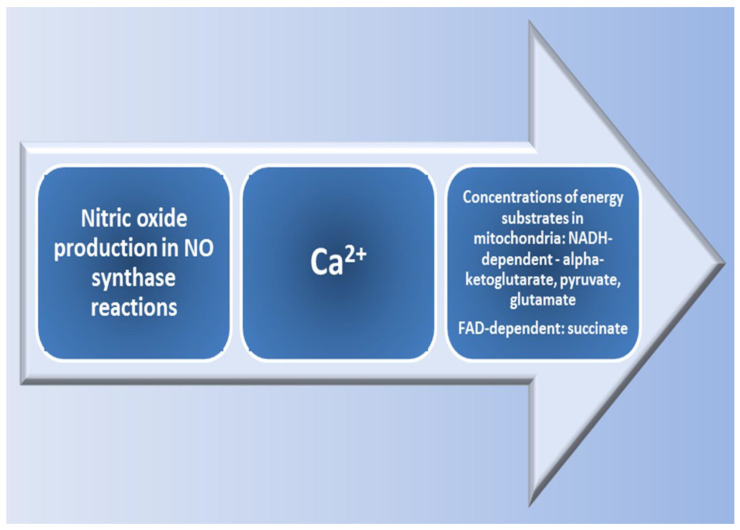
Role of calcium in the mitochondrial effects of energy support in hypoxia.

**Figure 5 ijms-24-08205-f005:**
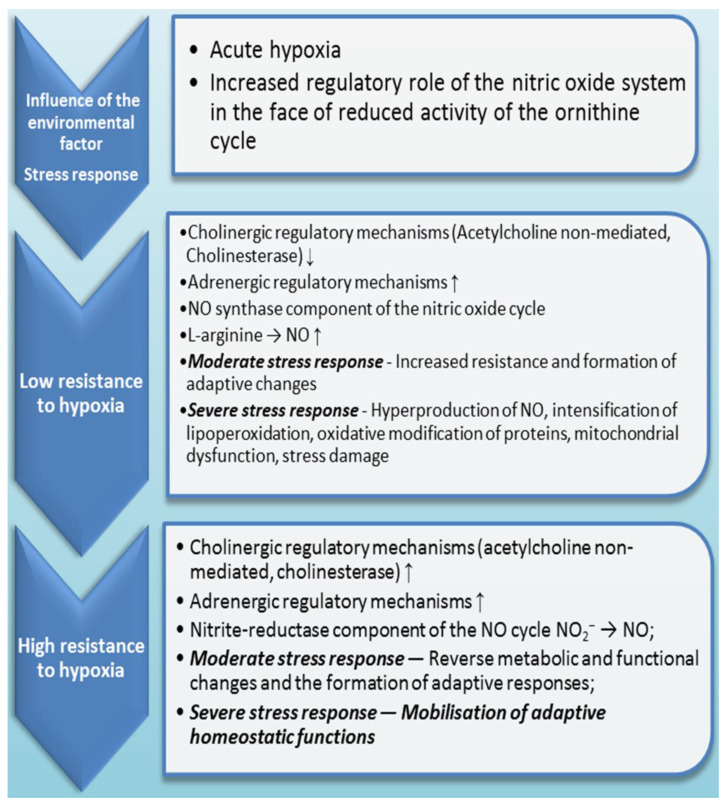
Involvement of NO^−^ synthase and nitrite-reductase components of the nitric oxide cycle in stress response depending on initial resistance to hypoxia.

**Figure 6 ijms-24-08205-f006:**
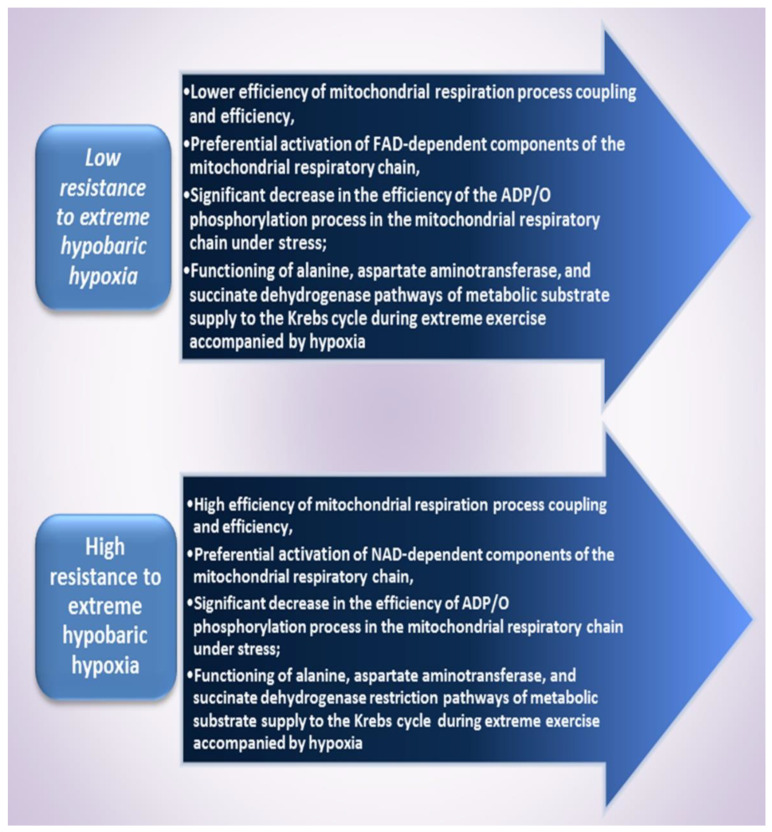
Cellular mechanisms of low and high resistance to hypoxia.

**Table 1 ijms-24-08205-t001:** Experimental models of L-arginine supplementation in therapeutic practice, effects, and proposed mechanisms of action.

N	Model	Effects of L-arginine	Mechanisms of L-arginine Action	References
1	Spontaneously hypertensive male rats SHR, Wistar Kyoto rats, L-arginine (10 g/L in drinking water), 1 week	Via local cardiac noradrenergic hyperactivity	Increased pre-synaptic substrate availability of the NOS-sGC-cGMP pathway reduced tyrosine hydroxylase levels	[28]
2	Spontaneously hypertensive young (12–14 weeks) and old (11–12 months) male Wistar rats	Aging effects similar to those seen in hypertension; age-dependent vascular dysfunction in SHRs is mediated by arginase	L-arginase reduces L-arginine availability for the formation of nitric oxide	[27]
3	Depletion of endogenous L-arginine due to maternal malaria infections by L-arginine or L-citrulline supplementation	Strategies for arginine supplementation in pregnancy	Implementation in resource-constrained settings, enhanced placental vascular development, and improved birth outcomes	[17]
4	Pregnant Female Sprague-Dawley rats, oral L-arginine supplementation from the 16th to 20th week with salt diets	Ameliorated deleterious effects in salt-induced hypertensive pregnant rats	NO vasodilatory effect	[18]
5	Pregnant patients with uteroplacental vascular dysfunction, high risk of adverse maternal and foetal outcomes	Pravastatin + L-arginine	Improved uteroplacental haemodynamics, increased foetal growth, and prevented early onset preeclampsia	[29]
6	Summarised data of preclinical and clinical studies of arginine/citrulline supplementation in adults and children	Improved endogenous NO regulation in cardiovascular diseases via endothelial function correction	Synthesis of NO from arginine/citrulline	[26]
7	Patients with pulmonary arterial hypertension	L-arginine and the L-arginine/ADMA ratio were lower in pulmonary arterial hypertension patients	L-arginine provided useful information in differentiating patients with pulmonary arterial hypertension diseases	[30]
8	Male and female rats with pulmonary arterial hypertension (monocrotaline sc, 60/kg b.b.) during nicotinamide (500 mg/kg, 7th day) and L-arginine supplementation (2.5% solution, drinking water, 7th day)	Protective effects on myocardial function and prevention of histopathological changes in pulmonary arteries	Clinical efficacy of supplementation via prevention of pulmonary arterial hypertension in a gender-dependent way	[31]
9	Patients with peripheral arterial disease and catheter-directed L-arginine delivery	Maximal effects for limb volumetric flow at 100 mg L-arginine supplementation	Correction of endothelial function in patients with peripheral arterial disease	[32]
10	Human lower extremity anterior tibial artery segments, histology and immunohistochemistry methods, amputation specimens in an ex vivo model	Improved endothelial dysfunction by L-arginine supplementation	Increase in the local levels of nitric oxide in humans and responsiveness to L-arginine as a nitric oxide precursor	[33]
11	Patients with fibrosis and severe secondary Raynaud’s phenomenon phenotypes at L-arginine-based therapies 1–2 g/day–10 g/day	NO metabolism in fibrosis and severe secondary Raynaud’s phenomenon phenotypes	eNOS, iNOS	[34]
12	Patients with primary or secondary Raynaud’s phenomenon at L-arginine-based therapy	Endothelial-derived mediators	Increased nitric oxide synthesis	[35]
13	Review of data from animal and clinical studies	Analysis of the states of hypertension, diabetes, hypercholesterolaemia, and vascular inflammation	L-arginine improved treatment of cardiovascular disorders via the NO pathway	[36]
14	Rats with high-fat diet and streptozotocin-induced hyperglycaemia, arginase inhibition (via L-norvaline) and supplementation (via L-arginine)	L-arginine acted as a potent antihyperglycaemic agent	Inhibition of arginase provided an antihyperglycaemic effect, NO protected against oxidative stress and hypercholesterolaemia	[37]
15	Patients with acute coronary syndrome, myocardial infarction and metabolic syndrome, L-arginine (4.2 g)/L-carnitine (2.0 g) in infusions for 28 days	Gradual recovery of myocardial contractility, reduction in diastolic dysfunction	Improved myocardial infarction protocol treatment in therapy of cardiovascular diseases	[38]
16	Subjects with early diabetes forms; arginine, ornithine, and citrulline level	Dipeptidylpeptidase-4 inhibitor linagliptin effects in subjects with coronary artery disease	No significant improvements in the arginine bioavailability ratios	[39]
17	Review of clinical and preclinical data in hypertension, ischaemic heart diseases, aging, peripheral artery disease, and diabetes mellitus; L-arginine supplementation	Nitric oxide synthase and arginase action, which are fundamental for the generation of NO	Supplementation of L-arginine prevented the evolution of hypertension and atherosclerosis	[40]
18	Patients with ischaemic heart failure, 3 g/d L-arginine supplementation, 10 weeks, cardiac reverse remodelling	Cardiac reverse remodelling after L-arginine supplementation	Improvement of cardiac recovery and function and quality of life in patients with ischaemic heart failure	[41]
19	Patients with type 2 diabetes mellitus different ages, oral supplementation with L-arginine 5 g/day for 14 days	No changes in glycaemia and lipidogram levels, decreased systolic, diastolic, and mean arterial pressure in elderly women, improved vasoreactivity	L-arginine as a precursor of NO synthesis improved endothelial-dependent vasodilatation and vascular/microvascular health in elderly women with or without type 2 diabetes mellitus	[42]
20	Older adult patients, short-term supplementation with l-citrulline (6 g day^−1^ for 14 days)	Modest improvement of muscle blood flow during submaximal exercise in older men	Possibility of L-citrulline to increase the L-arginine level	[43]
21	Glaucoma mouse models, precision glaucoma therapy, hydrophilic L-arginine	Intraocular pressure reduction	NO decreased high intraocular pressure	[44]
22	Wistar rats with myocardial infarction, L-arginine supplementation (1 g/kg, oral 7×/week), chronic heart failure, aerobic interval training	Interval training associated with L-arginine supplementation improved the hemodynamic parameters, reduction in pulmonary congestion	NO pathway improved the inflammatory profile and antioxidant status during training.	[45]
23	Wistar rats, chronic renal failure, L-arginine-NO system	Endothelial cell dysfunction at defective nitric oxide generation in chronic renal failure	Profound beneficial effects in chronic renal failure	[46]

**Table 2 ijms-24-08205-t002:** L-arginine and different hypoxia-induced animal models.

N	Model	Effects of L-arginine	Mechanisms of L-arginine Action	References
1	Murine cementoblast apoptosis and root resorption model, hypoxia-induced apoptosis, L-arginine application	Reduced cementoblast apoptosis and root resorption in hypoxia	Via improved Sirt1 activator resveratrol, activated autophagy in the root resorption model	[127]
2	Male Sprague-Dawley rats exposed to hypoxia, and hypoxia (altitude of 5000 m) plus a high-fat diet model, L-arginine supplementation for 1 week	Increased plasma nitrates and nitrites, endothelial nitric oxide synthase mRNA	Prevention of aortic ultrastructural changes via aortic endothelium effects and endothelium-dependent vasodilator response	[128]
3	Female and male Wistar rats high-fat diet model, L-arginine supplementation in a dose of 20 g/kg diet, vitamin C supplementation	Increased total antioxidant status, decreased insulin resistance, lowered LDL, reduced level of protein carbonyls	Via homocysteine levels, tumour necrosis factor alpha (TNF-α), oxidative stress biomarkers	[129]
4	Wistar rats with different resistance to hypoxia model, L-arginine single injection (600 mg/kg), blocker of nitric oxide synthase L-NNA (35 mg/kg) in single injection, liver tissue	ADP-dependent processes of oxidative phosphorylation with the use of different substrates of mitochondrial oxidation processes, calcium mitochondrial capacity, different mechanisms of the L-arginine impact depending on the individual resistance to hypoxia	Via activation of aminotransferase mechanism, ATP-dependent processes of oxidative phosphorylation, increases in mitochondrial calcium capacity in low resistant rats	[130]
5	Wistar rats intermittent hypoxic training model (11% O_2_, 15-min sessions with 15 min rest intervals, 5 times daily), acute hypoxia model (inhalation of 7% O_2_, 30 min), myocardium mitochondria, L-arginine impact, NO blocker, L-NNA impact	ADP-dependent processes of oxidative phosphorylation with the use of different substrates of mitochondrial oxidation processes, increase in the tolerance to episodes of acute hypoxia	Stimulation of oxidative phosphorylation with primary activation of NAD-dependent mitochondrial pathway, a marked increase in the ADP/O ratio	[131]
6	Coronavirus disease (COVID-19) epidemiological situation, L-arginine supplementation, stress conditions, pulmonary diseases	May attenuate SARS-CoV-2 infection	Via restoration of NO by L-arginine antiviral and immunomodulatory effects, reduction of binding of SARS-CoV-2 to angiotensin-converting enzyme 2, inhibition of transmembrane protease serine-type 2, inhibition of proliferation and replication of SARS-CoV-2	[132]
7	Seven-day-old rat hypoxia-ischemia model, L-arginine impact before hypoxia-ischemia, analysis of neuronal apoptosis biomarkers with the dUDP-biotin nick end-labelling (TUNEL) method, apoptosis indexes of the hippocampus and striatum	No significant difference in the right apoptosis indexes of the cortex	Via NO production induced by L-arginine post-treatment	[133]
8	Hypoxia induced by sodium nitrite (75 mg/kg s.c) neurotoxicity rat model, L-arginine, and carnosine impact	Combination of L-arginine and carnosine protects against hypoxia-induced neurotoxicity	Via inflammatory mediators, including nuclear factor kappa B, angiogenic, anti-inflammatory, and anti-apoptotic properties via tumour necrosis factor-alpha, caspase-3, GABA, noradrenaline, serotonin	[134]
9	Fish aquaculture system model, fish *Cirrhinus mrigala* exposed to hypoxia animal model, dietary arginine effects (0.7 and 1.4%) during 60 days	Effective supplement for aquaculture to reduce metabolic changes caused by hypoxia through increased antioxidant protection	Via hypoxia inducible factor (HIF)-1α mRNA	[135]
10	Myocardial infarction-induced damage rat model, L-arginine treatment in drinking water (4 g/L), high-intensity interval training alone and synergistically with L-arginine effects model, ischemia-reperfusion injury model	Synergic effects in improved left ventricular function at ischemia-reperfusion, oxidative stress mitigation, angiogenesis amelioration	Via cardiac function, angiogenesis, oxidative stress, and infarction size effects	[136]
11	T-mTNPs@L-Arg multifunctional nanoplatform as mitochondrial targeting nitric oxide model, long-term hypoxia breast cancer model, NO gas therapy model	Synergistic strategy in breast cancer	Effective proposal for nitric oxide gas mitochondrial targeting therapy in cancer	[137]
12	In vivo rat models at 14-day analysis of progression and treatment of pulmonary hypertension, hypoxic chamber (10% O_2_) method, different NO donor methods, l-arginine (20 mg/mL), molsidomine (15 mg/kg in drinking water)	Similar effects of different NO donors in enhancing the NO-cGMP pathway in pulmonary hypertension processes	Via modulation of the NO-cGMP pathway in vivo	[138]
13	Mouse renal epithelial cells in hypoxic conditions model, *Arg-II*^*−/−*^ mice model; cultured human renal epithelial cell line HK2 model	Effects caused by hypoxia-mediated metabolism of L-arginine into urea and L-ornithine pathway	HIFs-Arg-II-mtROS-TGFβ1-cascade, type-II arginase, hypoxia-inducible factors HIF1α and HIF2α	[139]
14	Hypoxia-induced hypertensive rat model	Inhibitory effect on eNOS and activating effect on arginase II and nitric oxide bioavailability	Via nitric oxide concentrations, dimethylarginine dimethylaminohydrolase-2, cystathionine β-synthase, homocysteine	[140]
15	Adult Wistar rats at chronic intermittent hypoxia (2 days of hypoxia/2 days of normoxia) and chronic hypoxia in whole lung tissue models	Decreased L-arginine/ADMA ratio	Via NO bioavailability changes, asymmetric dimethylarginine increase, oxidative stress biomarker malondialdehyde rise	[141]

## Data Availability

Not applicable.

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
