# Peer review of "The Effectiveness of L-arginine in Clinical Conditions Associated with Hypoxia"

_ijms, 2023, doi:10.3390/ijms24098205_

Round 1
Reviewer 1 Report
The manuscript is focused on THE EFFECTIVENESS OF L-ARGININE IN CLINICAL CONDITIONS ASSOCIATED WITH HYPOXIA. The present manuscript has been written well but I have some queries?
1. No material and method section?
2. From where and how they collect the data no information? Which is the most important factor for review as author mention last 50 years data.
3. Most of the references are from 2015 to 2023.
4. Author should also add figure to represents: various levels of regulation of NOS mRNA expression, posttranslational protein modification, and catalytic function. And Mode of action L-Arginine.
The data not well organized.
5. Conclusion should be highlight the importance of current reviews and future prospective.
6. Typographical mistakes check throughout the manuscript.
Author Response
Dear Prof. the Editor,
I thank you and the anonymous reviewers for your excellent feedback to my manuscript and for your constructive comments. These comments are very important for improving the quality of my manuscript and for my follow-up studies. I have extensively revised my manuscript based on your important comments.
Reviewer #1: Comment 1: No material and method section?
Response 1: Thank you very much for your positive evaluation and constructive suggestions. According to the review article format concept and editorial requirements, the review may not include a Materials and Methods section, so I did not do so.
Comment 2: From where and how they collect the data no information? Which is the most important factor for review as author mention last 50 years data.
Response 2: Thank you very much for your positive evaluation and constructive suggestions. The topics presented in the review are constantly topical and are actively discussed in the various sources available to the author (internet, digital and non-digital data, etc.) for the last 50 years. The author has been following these sources for a long time. The scope of this work does not allow to present all sources in a one review, as the editors and one of the reviewers required that more than 50% of the references in this review be cited in the last 5 years. It should be noted that in PubMed there are thousands of citations for the key words nitric oxide and L-arginine, and the author's task is to present his original concept of the paper. Therefore, this limits the authors' possibilities. This forces authors to write new articles on similar topics, which will be done by the author in the future, taking into account the reviewer's suggestions.
Comment 3: Most of the references are from 2015 to 2023.
Response 3: Thank you very much for your positive evaluation and constructive suggestions. The topic of a journal article in the review form has its own limitations for authors. One of these limitations was the requirement imposed by the editorial board and one of the reviewers that more than 50% of the papers in this review must have been cited in the last 5 years. However, the topic under review covers a much wider spectrum of time.
Comment 4: Author should also add figure to represents: various levels of regulation of NOS mRNA expression, posttranslational protein modification, and catalytic function. And Mode of action L-Arginine.
Response 4: Thank you very much for your constructive comments. The reviewer's comments are very constructive and important, but they can form the basis for another, with a different research topic of a review or research paper. We will certainly take this into account in the future when writing new articles on these topics. We have based our research on the topic of the effect of L-arginine in the clinical practice of hypoxia and have already used two large tables and 6 figures for this. The possibility of publishing articles in a journal in this situation is presented in terms of significantly less number of tables and figures. Unfortunately, the author has to take into account all comments of both the editorial board and the reviewers, as well as the journal conditions.
Comment 5: The data not well organized.
Response 5: Thank you very much for your comments to our manuscript. The reviewer's comments are very constructive and important, but the reviewer did not indicate which data are not well organised. This makes it difficult for the author to correct it. Previously the reviewer
wrote that the work is well written, and this is also noted by the other two reviewers. This situation, in which the author has to take into account the whole complex of all the comments to the paper in correction, puts the author in a difficult situation. Thanks to the reviewer for this comment, however, we will work on it.
Comment 6: Conclusion should be highlight the importance of current reviews and future prospective.
Response 6: Thank you very much for your constructive comments. Accordingly, the following comments, marked in a different colour, have been added to the text of the Conclusions section.
Therefore, the importance of this review for future perspectives on the methodological challenges of physiological experimentation and the therapeutic implications of treatment consists in including a component that depends on the baseline level of physiological reactivity.
For details, please see the revised manuscript.
Comment 7: Typographical mistakes check throughout the manuscript.
Response 7: Thank you very much for your positive evaluation and constructive suggestions. A professional proofreading was done for this work by experts, so I would be very grateful to the reviewer for pointing out specific positions in the manuscript text with these errors. It would be a great help to me to go back to these professionals for additional verification of the reviewer's recommendations.
Thank you very much for your constructive comments.
Author.

Reviewer 2 Report
In the article entitled "The Effectiveness of L-Arginine in Clinical Conditions Associated with Hypoxia", the authors do a very complete review of arginine when the person or organism is under hypoxic conditions. The manuscript is well presented and with enough schemes to make it quite understandable. There is also a table on experimental models of L-arginine supplementation as a therapeutic practice, as well as the mechanisms of action.
From my point of view the article is suitable for publication in the Journal.
Author Response
Dear Prof. the Editor,
I thank you and the anonymous reviewers for your excellent feedback to my manuscript and for your constructive comments. These comments are very important for improving the quality of my manuscript and for my follow-up studies. I have extensively revised my manuscript based on your important comments.
Reviewer #2: The suggestion from Reviewer 2: In the article entitled "The Effectiveness of L-Arginine in Clinical Conditions Associated with Hypoxia", the authors do a very complete review of arginine when the person or organism is under hypoxic conditions. The manuscript is well presented and with enough schemes to make it quite understandable. There is also a table on experimental models of L-arginine supplementation as a therapeutic practice, as well as the mechanisms of action.
From my point of view the article is suitable for publication in the Journal.
Comment 1: Thank you very much for your positive evaluation and constructive suggestions.
Thank you very much for your constructive comments.
Author.

Reviewer 3 Report
A very comprehensive review of the use of l-arginine in therapeutic regimens, which is structured well and gives strong evidence to support it's conclusions.
Whilst the text is good with only a few minor language errors, the quality of figures and tables could be significantly improved. Some of the text in very dense on some of the figures and it looks untidy with inconsistencies in font sizes. Also figure and table legends lack detail and should be more descriptive.
Just a few errors in sentence construction in the text, e.g. li321 '...by used athletes...'
Author Response
Dear Prof. the Editor,
I thank you and the anonymous reviewers for your excellent feedback to my manuscript and for your constructive comments. These comments are very important for improving the quality of my manuscript and for my follow-up studies. I have extensively revised my manuscript based on your important comments.
Reviewer #3:
Comment 1: The suggestion from Reviewer 3: A very comprehensive review of the use of l-arginine in therapeutic regimens, which is structured well and gives strong evidence to support it's conclusions. Whilst the text is good with only a few minor language errors, the quality of figures and tables could be significantly improved. Some of the text in very dense on some of the figures and it looks untidy within consistencies in font sizes. Also figure and table legends lack detail and should be more descriptive.
Response 1: Thank you very much for your reminder. This is a good question. The reviewer may have been looking at the first version of the article, which was later significantly modified according to the instructions of other reviewers. Accordingly, the manuscript text has been changed and Еable 2 has been added. Also the figures have been changed. According to the reviewers' requirements, the figures themselves should not duplicate the text and vice versa. I have followed these requirements of the reviewers by composing the information presented in the paper. I would be grateful to the reviewer for pointing out specific errors in the text. This is important because the text has been previously handled by a professional proofreader in order to send him comments on suggested language errors.
For details, please see the revised manuscript.
Thank you very much for the positive evaluation and constructive suggestions.
Author

Round 2
Reviewer 1 Report
Author responded all the queries.